# Altered chemistry of oxygen and iron under deep Earth conditions

Jin Liu [1,2], Qingyang Hu [1], Wenli Bi[3,4], Liuxiang Yang [1,5], Yuming Xiao[6], Paul Chow[6], Yue Meng[6]
Vitali B. Prakapenka[7], Ho-Kwang Mao[1,5] & Wendy L. Mao[2,8]

A drastically altered chemistry was recently discovered in the Fe-O-H system under deep Earth conditions, involving the formation of iron superoxide ($FeO_2Hx$ with $x = 0$ to 1), but the puzzling crystal chemistry of this system at high pressures is largely unknown. Here we present evidence that despite the high O/Fe ratio in $FeO_2Hx$, iron remains in the ferrous, spin-paired and non-magnetic state at 60–133 GPa, while the presence of hydrogen has minimal effects on the valence of iron. The reduced iron is accompanied by oxidized oxygen due to oxygen-oxygen interactions. The valence of oxygen is not –2 as in all other major mantle minerals, instead it varies around –1. This result indicates that like iron, oxygen may have multiple valence states in our planet's interior. Our study suggests a possible change in the chemical paradigm of how oxygen, iron, and hydrogen behave under deep Earth conditions.

[1] Center for High Pressure Science and Technology Advanced Research, Beijing 100094, China. [2] Department of Geological Sciences, Stanford University, Stanford, CA 94305, USA. [3] Advanced Photon Source, Argonne National Laboratory, Argonne, IL 60439, USA. [4] Department of Geology, University of Illinois at Urbana-Champaign, Urbana, IL 61801, USA. [5] Geophysical Laboratory, Carnegie Institution of Washington, Washington, DC 20015, USA. [6] HPCAT, X-Ray Science Division, Argonne National Laboratory, Argonne, IL 60439, USA. [7] Center for Advanced Radiation Sources, University of Chicago, Chicago, IL 60439, USA. [8] Stanford Institute for Materials and Energy Sciences, SLAC National Accelerator Laboratory, Menlo Park, CA 94025, USA. Correspondence and requests for materials should be addressed to Q.H. (email: qingyang.hu@hpstar.ac.cn) or to H.-K.M. (email: hmao@gl.ciw.edu) or to W.L.M. (email: wmao@stanford.edu)

Oxygen and iron are Earth's most abundant elements by number of atoms and by mass, respectively. They form compounds dictating major chemistry of our planet[1]. It is conventionally accepted that the oxygen anion has an unvarying $-2$ valence state in mantle ferropericlase and bridgmanite throughout the deep interior, where the oxygen fugacity decreases with increasing depth. The redox states are mostly controlled by the $3d$ transition element Fe which could vary among three valence states, metallic $Fe^0$, ferrous $Fe^{2+}$, and ferric $Fe^{3+}$. Recently, a series of new iron oxides have been found with varying O/Fe stoichiometry ranging from the end-member $Fe^{3+}_2O_3$ on our planet's highly oxidized surface to the other end-member $Fe^{2+}O$ which should be stable at the highly reduced conditions in the deep lower mantle down to the core-mantle boundary, which include $Fe_5O_7$, $Fe_4O_5$, and $Fe_5O_6$ (refs. [2–4]). High pressures in the deep lower mantle would promote the crystal field splitting of $3d$ orbitals of iron and cause the electronic spin-pairing transition, which can affect the physical, chemical, and transport properties of mantle phases[5–7].

The conventional wisdom, however, is facing a change in light of the recent discovery of the high-pressure pyrite-structured iron superoxide $FeO_2$ which has O/Fe ratio even higher than $Fe_2O_3$ and can hold a varying amount of hydrogen (denoted as "Py-$FeO_2Hx$" with $x$ from 0 to 1)[8–12,13–15]. With the subducted plate carrying down water to react with the iron core to form Py-$FeO_2Hx$ and release hydrogen, oxygen-rich reservoirs could be accumulated in the very reducing core-mantle boundary region[13–15]. Such reservoirs at the mid-point (2900 km depth) of the Earth's radius (6370 km) will certainly play a pivotal role in the global chemistry, including the generation of our present day aerobic atomosphere[9,16]. A number of key solid-state chemistry questions on Py-$FeO_2Hx$ must be understood: What are the valence states of Fe and O? What is the nature of their chemical bonding? What are the effects of hydrogen on the valence and bonding of Fe and O? Is the iron superoxide metallic? Are its $d$-electrons spin-paired?

Assuming oxygen in the common $O^{2-}$ state as in other iron oxides, the valence state of iron would be ferryl ($Fe^{4+}$) for $FeO_2$. On the other hand based on the analogy to $FeS_2$ pyrite, in which Fe remains ferrous ($Fe^{2+}$) with the sulfur forming $S_2^{2-}$ dimer, would $FeO_2$ also consist of $Fe^{2+}$ cations and $O_2^{2-}$ dimer? Streltsov et al.[17] predicted the valence state of iron to be $+3$ for the hydrogen-free $FeO_2$ Py-phase where oxygen ions do not form $O_2^{2-}$ dimers, suggesting $FeO_2$ "in between" the usual dioxides and peroxides, based on the density functional and dynamical mean-field theories (DFT + DMFT). By contrast, Jang et al.[18] employed DFT + DMFT approaches with treating $FeO_2$ to presumably have $O_2^{2-}$ dimer as a complete analogue of pyrite-structured $FeS_2$. Thus far, theoretical calculations cannot conclude the nature of chemical bonding and state of $FeO_2$ without experimental constraints. Answers must be based on in situ measurements of the micron-sized Py-$FeO_2Hx$ sample which cannot be preserved after releasing pressure for normal electrochemical analysis.

We take the challenge by combining a variety of recently enabled synchrotron X-ray spectroscopic techniques[19], and obtain intriguing answers from direct high-pressure experimentation supplemented by first-principles simulations with the self-consistent linear-response approach. Here we find many unexpected chemical behaviors in Py-$FeO_2Hx$ samples (Supplementary Figures 1–5), where iron is in the reduced, spin-paired ferrous state and the valence state of oxygen varies around $-1$, instead of at $-2$ in common iron oxides, while hydrogen is weakly bonded in the structure. Our results suggest that other than iron, oxygen can occupy multiple valence states in our planet's interior, leading to a chemical paradigm change in the deep Earth.

## Results

**Valence states of iron and oxygen in $FeO_2Hx$.** The valence state of Fe can be characterized by near-edge X-ray absorption spectroscopy (XAS) features at energy slightly below the iron $K$-edge[20]. Because of the strong absorption of 7 keV X-ray by diamond anvils, the XAS spectra were collected in the partial fluorescence yield (PFY) geometry for the iron superoxide at 53–133 GPa. That is, the incident X-ray beam passes through one diamond anvil, and then the X-ray fluorescence exits in the radial direction through the Be gasket to avoid double diamond absorption. The peak position and intensity distribution are diagnostic probes for valence state, spin state, geometry, and the ligand–ligand interactions for Fe atoms[20]. The iron $K$ pre-edge feature that is related to the $1s \rightarrow 3d$ electronic transition, is particularly sensitive to the valence state of iron[21]. Our XAS-PFY measurements of other representative iron compounds at ambient conditions explicitly show that the position of iron $K$ pre-edge absorption increases with increasing valence state (Fig. 1a top). The position of iron $K$ pre-edge of the iron superoxide $FeO_2$ is very close, but slightly below, that of $Fe^{2+}$ compounds. That is, the valence state of iron in $FeO_2$ is plausibly close to $+2$. Its position remains almost constant from 53 to 81 GPa, suggesting a minimal pressure dependence on the valence state of iron[22], consistent with mantle ferropericlase and bridgmanite. Furthermore, it is intriguing that the iron valence state of the hydrogen-free iron superoxide is similar to that of the hydrogen-bearing Py-$FeO_2Hx$ with $x = 0.5$–$0.7$ at 133 GPa. These observations indicate that the hydrogen content has minimal effects on the valence state of iron in the superoxide (Fig. 1a).

With the valence state of iron slightly under $+2$, the valence state of oxygen for the Py-$FeO_2$ would have to be nearly $-1$, suggesting that oxygen atoms must interact with each other in the iron superoxide. Such interaction can be observed experimentally from the oxygen $K$-edge spectroscopy. We note that the edge at 540 eV is in a very soft X-ray region inaccessible to XAS in DAC experiments. Instead, we can access the oxygen $K$-edge by X-ray Raman scattering (XRS) using hard X-ray in and hard X-ray out method, where the energy loss is measured due to inelastic Raman shift[23,24]. For XRS measurements on the oxygen $K$-edge, we deliberately used LiF as a pressure transmitting-medium, instead of water or $O_2$, to assure XRS signals of oxygen only from the Py-$FeO_2Hx$ (Supplementary Figure 3). In the XRS at 110 GPa, we observed a sharp oxygen $\pi^*$ peak near 531 eV and a broad $\sigma_u^*$ peak at 540–550 eV (Fig. 1b). These peaks represent the O electron transition from $1s$ core orbitals to the antibonding $1\pi_g^*$ and $3\sigma_u^*$ orbitals, respectively (termed as $\pi^*$ and $\sigma^*$ transitions). The strong $\pi^*$ transition indicates the presence of the O–O dimer.

**Magnetic and electronic spin states of iron in $FeO_2Hx$.** Magnetic ordering and spin state of Fe in iron superoxide can be investigated using synchrotron Mössbauer scattering (SMS) and X-ray emission spectroscopy (XES), respectively. $^{57}Fe$ is a Mössbauer active isotope with nuclear resonant energy of 14.41 keV. A series of SMS spectra of the iron superoxide were collected upon decompression in steps from 118 GPa down to 32 GPa (Fig. 2a), and the experiment was repeated in three separated runs (Supplementary Figures 6–8) of $FeO_2Hx$ with different $x$ (Supplementary Table 1). SMS spectra were analyzed using CONUSS program[25] to extract the quadrupole splitting (QS) and isomer shift (IS) of iron in the Py-phase which is compared with that of standard samples of ferrous pyrite $FeS_2$, ferrous $Fe_{0.48}Mg_{0.52}O$, and ferric $Fe_2O_3$ (Fig. 2b). The QS of iron in the Py-phase barely

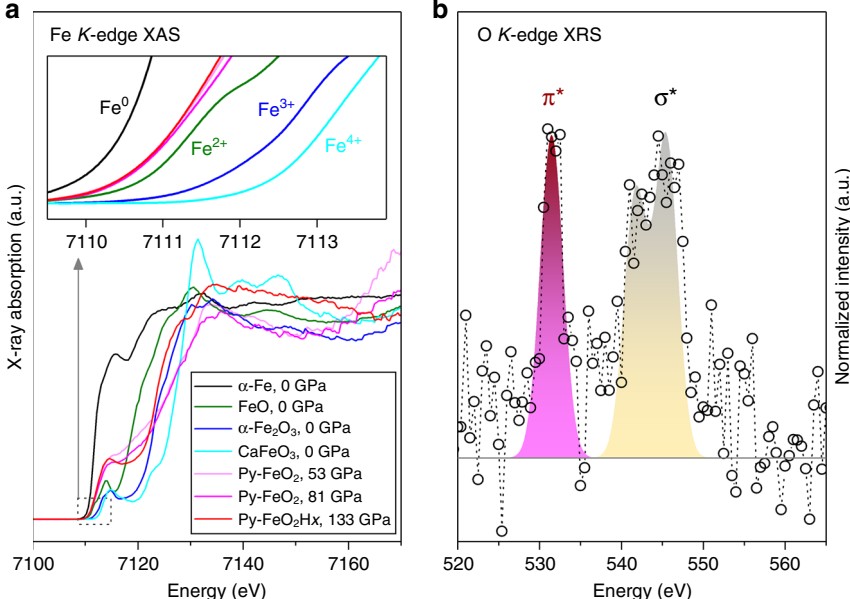

**Fig. 1** Representative XAS and XRS spectra of iron oxide compounds. **a** PFY-XAS spectra at Fe $K$-edge of iron oxide compounds at room temperature. Black, olive, blue, and cyan lines: $Fe^0$, $Fe^{2+}O$, $Fe^{3+}_2O_3$, and $CaFe^{4+}O_3$ at ambient conditions, respectively; light and dark magenta lines: Py-FeO₂ at 53 and 81 GPa, respectively; red line: Py-FeO₂H$x$ at 133 GPa; gray arrow: the link from the dashed outline to the inset. Inset: the area zoomed for the dashed outline in Fig. 1a. **b** XRS spectra of Py-FeO₂H$x$ at 110 GPa. Circles: experimental data; shaded areas: fitted peaks

increased with increasing pressure, whereas the IS displayed an opposite trend. The pressure dependence of QS and IS for iron in the superoxide is comparable to that for ferrous iron ($Fe^{2+}$) in ferropericlase (Mg, Fe)O in the low-spin state (refs. [26,27]), likely due to the ferrous nature of the iron superoxide. It is noted that the spin transition could largely affect the QS of iron in (Mg, Fe)O (refs. [27,28]). Furthermore, the values of the QS and IS of iron in the superoxide extrapolated to ambient conditions are about 0.27 and 0.55 mm s$^{-1}$, in excellent agreement with that in pyrite FeS₂ (ref. [29]).

Our SMS spectra in Fig. 2a consistently showed that the Py-phase remained in the non-magnetic state represented by the clean, single beat spectra, regardless of the content of hydrogen in the lattice. SMS data with multiple time beats indicating magnetic ordering appeared only below 40–50 GPa when the iron super-oxide decomposed into a magnetic phase under decompression. We note that the full width at half maximum (FWHM) for the diffraction peaks for the iron superoxide broadened significantly below 65 GPa and the peak intensity continually decreased until they finally disappeared around 40 GPa upon decompression at room temperature. Furthermore, the QS and IS values of iron in the magnetic phase are close to that in hematite ($Fe_2O_3$) at 32 GPa. It is consistent with our recent X-ray diffraction (XRD) observations where the new diffraction peaks can be indexed to hematite below 40 GPa.

The magnetic spin states of $3d$ electrons are also directly probed by XES spectra of the Fe $K_\beta$ fluorescence lines[5,30] of the iron superoxide between 40 and 133 GPa (Fig. 3). The intensity of the $K'_\beta$ satellite peak of iron in the superoxide decreased between 45 and 60 GPa and disappeared completely at 60–65 GPa where the total spin momentum ($S$) of iron in the Py-phase approached zero, indicating that iron in the Py-phase undergoes a gradual electronic spin-pairing transition below 65 GPa from high-spin to low-spin states (Supplementary Figure 9). We note that the XES spectra were collected with decreasing pressure and thus the spin transition pressure may be higher due to pressure hysteresis on the spin crossover upon decompression[31]. These observations

also suggest that the iron superoxide is not stable in the high-spin state and would decompose into oxygen and hematite across the electronic spin transition of iron, associated with the valence change in oxygen and iron at the same time.

**Oxygen bonding and charge transfer in FeO₂ and FeO₂H.** For pyrite-type structures, the covalent nature is mainly controlled by the anion–anion distance ($D_O$). We followed previous strategies[32,33] and used the fractional coordination $y$ of oxygen to compare O–O distance at different pressures:

$$y = \frac{1}{2} - \frac{\sqrt{3}D_0}{6a}, \tag{1}$$

where $a$ is the lattice parameter as a function of pressure. The archetypical pyrite FeS₂ with $y = 0.385$ forms solid anion dimers with divalent Fe cations at ambient conditions. In contrast, conventional dioxides with tetravalent cations (e.g. RuO₂ and OsO₂) have a smaller value of 0.34–0.35 for $y$ in the pyrite-type structure[33]. Our previous experimental XRD data showed $y = 0.371$ with $D_0 = 1.937$ Å in the iron superoxide at 75 GPa which lies between conventional dioxides and peroxides[9,33]. Due to the importance of the O–O bond length, we collected XRD patterns on a nearly pure FeO₂ sample at 96 GPa, which have been further analyzed by using the Rietveld method for the primary phase of FeO₂ and the LeBail method for the weaker O₂ and Fe₂O₃ phases. The final refinement factors are $R_1 = 0.069$ and $wR_2 = 0.14$. The bond length ($D_0$) for FeO₂ is ~1.87(1) Å and the lattice parameter is 4.2855(3) Å with $y = 0.374$ at 96 GPa. Therefore, compared with various peroxides synthesized at ambient pressure[34], the O–O bonding length in the iron superoxide is relatively longer. With $0.35 < y < 0.39$, iron–oxygen octahedrons are chained by the O–O bonding as shown in Fig. 4a.

We further conducted first-principles simulations (see Methods) to construct the charge density landscape at 100 GPa (Fig. 4). Following the Bader charge division scheme, we determined the charge of the Fe atoms are 6.38 e by DFT + U and 6.32 e by the

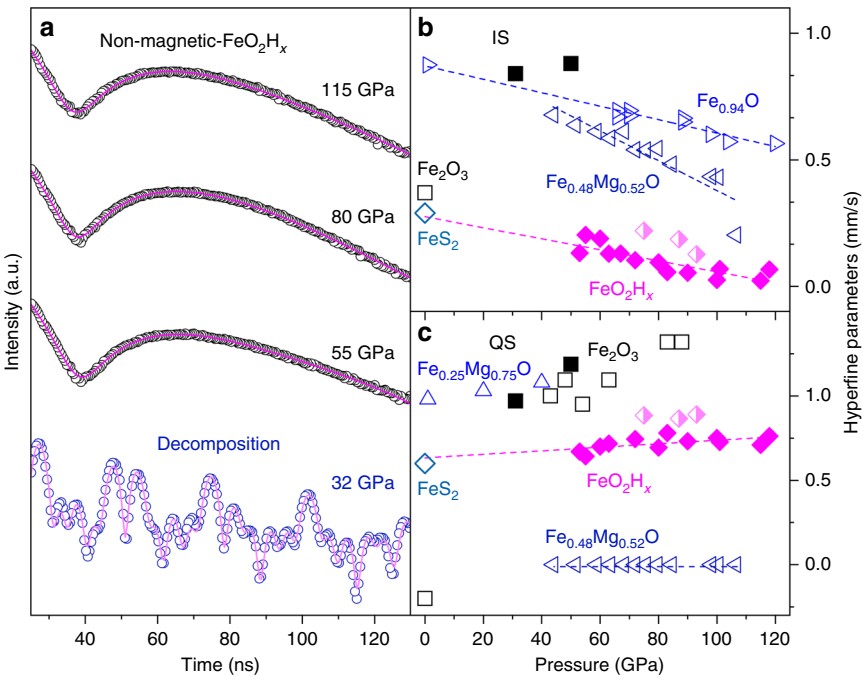

**Fig. 2** Representative SMS spectra and hyperfine parameters. **a** High pressure Mössbauer spectra of $FeO_2Hx$ collected upon decompression at 300 K. Circles: experimental SMS spectra; red solid lines: modeled spectra using CONUSS program. Comparison of isomer shift (**b**) and quadrupole splitting (**c**) of iron in iron-rich compounds at high pressure and room temperature. Solid and half-filled diamonds: pyrite-structured $FeO_2Hx$ and $FeO_2$, respectively, this study; open diamonds: pyrite $FeS_2$ (ref. [29]); solid squares: $Fe_2O_3$, this study; open squares: $Fe_2O_3$ (ref. [58]); left-, right-, and top-pointing triangles: ($Fe_{0.48}Mg_{0.52}$)O (ref. [27]), $Fe_{0.94}O$ (ref. [26]), and ($Fe_{0.25}Mg_{0.75}$)O (ref. [28]), respectively; dashed lines: linear fits to experimental data. The errors on the experimental data are ±2 SD, which are smaller than symbols and not shown for clarity

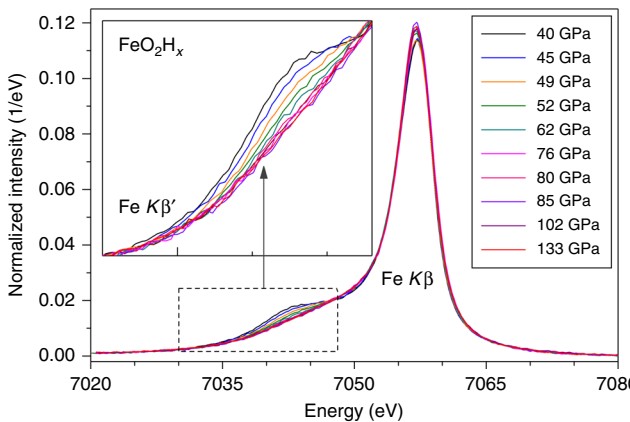

**Fig. 3** High-pressure XES spectra of pyrite-structured $FeO_2Hx$ at 300 K. The integrated intensity of the XES spectra was normalized to unity. Inset: the $K'_\beta$ satellite emission peak between 7030 and 7048 eV after the iron main emission peaks was aligned to that at 133 GPa. The changes of the satellite peak intensity in $FeO_2Hx$ are attributed to the high-spin to low-spin transition, as the disappearance of the satellite peak has been used as a robust criterion for the electronic spin-pairing transition of iron in other iron-bearing compounds[30]

Heyd-Scuseria-Ernzerhof (HSE) screened hybrid functional (screening variable of 0.15) free of the empirical Hubbard parameter $U$. Based on our experimental observations that Py-$FeO_2Hx$ is in the non-magnetic state, the O–O bond length values from our GGA + $U$ calculations are consistent with our XRD experimental results as well as Lu et al.[35]. We note that the O–O bonding length is sensitive to computational and physical

environments and that either antiferromagnetic or ferromagnetic configurations would yield a longer O–O bond by ~10%[35]. The valence state of iron is +1.62 (DFT + $U$) or +1.68 (HSE) and that of oxygen is −0.81 (DFT + $U$) or −0.84 (HSE) for $FeO_2$ (Table 1). It is known that Bader analysis may underestimate the charge transfer between anions and cations[36]. Our calculations may not indicate the exact valence values, but clearly show charge transfer between anions, which has also been seen in the calculations of archetypal pyrite $FeS_2$ (ref. [37]). In hydrogen-bearing Py-$FeO_2H$, hydrogen is equally bonded with two nearest oxygen atoms[38]. As a result, hydrogen is calculated to be +0.64 at 100 GPa. Oxygen atoms adopt electrons from hydrogen and become −1.13 while the valence state of iron remains around +1.63 for the Py-$FeO_2H$. Therefore, hydrogen only donates electrons to oxygen atoms and has a minimal effect on the valence state of iron.

## Discussion

Looking at individual properties, $FeO_2$ and $FeO_2Hx$ may appear similar to pyrite $FeS_2$ in their crystal structure, cation and anion valences, and spin-pairing state. Assimilating the comprehensive experimental and theoretical investigations, however, points to fascinating changes of high-pressure crystal chemistry above 70 GPa which corresponds to roughly the pressures beyond 1700 km depth, which is the midpoint separating the Earth's mass into two equal halves. At higher pressures, the valence of oxygen in oxides is no longer fixed to −2, but becomes variable. The valence of iron becomes more dependent on the crystal structure and pressure rather than the oxygen stoichiometry. Hydrogen is no longer bonded to oxygen as OH, but becomes mobile. $FeO_2$ and $FeO_2Hx$ do not exactly belong to the pyrite $FeS_2$ type structure with interactive anion dimer, nor the $PdF_2$ type structure without interactive anion dimer, but represent a new structure type in the $Pa\bar{3}$ $MA_2$ family with

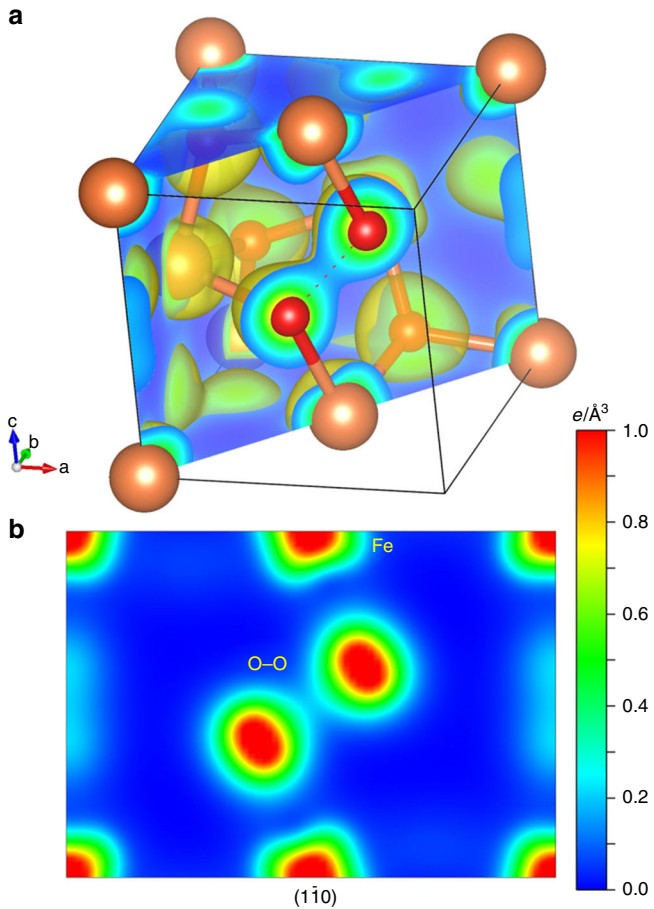

**Table 1 The valence state of Fe and O atoms from the electron charge density of Py-phase at 100 GPa and 0 K**

| FeO$_2$ | DFT + U | HSE |
|---|---|---|
| Fe | 1.62 | 1.68 |
| O | −0.81 | −0.84 |
| $y$ | 0.367 | 0.362 |

**Fig. 4** The calculated charge density landscape of pyrite-structured FeO$_2$ at 100 GPa. **a** Gold surface represents charge density of 0.15 e/Å$^3$. The section cut through the (1$\bar{1}$0) plane shows the weak O–O bonding feature. Brown balls are Fe atoms and red balls are O atoms. **b** Charge density distribution projected on the (1$\bar{1}$0) plane. The level of charge density is reflected by the color bar

partially interactive anion dimer. These issues have been previously noted as separate enigmas in other studies of O, Fe, and H under high pressures, such as formation of O$_8$ molecules[39], the redox paradox of ferric iron[40], and mobility of hydrogen in ice[41,42]. With the integrated study here, now the overall picture clearly indicates a chemical paradigm change under deep Earth conditions.

At zero pressure without external constraint, the O–O interaction type can be empirically determined by their distance[33,34], 1.29–1.53 Å for bonded dimers in peroxides and 2.45–2.72 Å for non-interactive oxygen in PdF$_2$-type dioxides. Under pressures, however, the intermolecular interactions are further controlled by additional external forces that can generate additional intermolecular bonds. For instance, O$_2$ transform to the high pressure ε-O$_8$ phase with two types of bond length $D_0 = 1.20$ Å and 2.18 Å, respectively. The O–O dimer of $D_0 = 1.89$ Å (refs. [24,39,43]), the reduced valence of oxygen in FeO$_2$ and FeO$_2$H$x$, and the formation of new type of $Pa\bar{3}$ peroxide are a natural consequence of high-pressure chemistry.

At zero pressure, the valence of iron or ferric/ferrous ratio is generally correlated to the oxygen fugacity. The correlation is weakened by the additional control by the crystal structure of high-pressure phases. For instance, the paradoxical correlation of lower oxygen fugacity with higher Fe$^{3+}$ abundance[40] reflects the ability of the high-pressure perovskite structure to

accommodate more Fe$^{3+}$ iron[44]. Here the new high-pressure iron superoxide structure is able to keep its iron in ferrous state by reducing the oxygen valence, and fits well in the reducing environment of the core-mantle boundary. The spin-paired non-magnetic Fe$^{2+}$ also fits the high-pressure chemistry of the deep lower mantle[5,30].

At zero pressure, hydrogen is present as H$^+$ and replaces other cations without affecting the anion valence. Here, the addition of hydrogen has negligible effects on the chemical and magnetic properties of the iron in FeO$_2$H$x$, but mainly affects the oxygen valence. This shows the change of hydrogen from strong OH covalent bond at low pressures to weak interaction in the Py-FeO$_2$H$x$ lattice. Hydrogen chemistry in H$_2$O is known to have a very important change at high pressures to become symmetrically hydrogen bonded at low temperature[42] and superionic at high temperature[41,45]; both result in weakening the OH bonding. The present observation indicates the weakened interaction of hydrogen is not limited to H$_2$O but is a general high-pressure chemical trend that has many important consequences, including loss of hydrogen in the formation of Py-FeO$_2$H$x$ (ref. [38]), spontaneous splitting of H$_2$O into hydrogen and oxygen[13], and separation of water and hydrogen cycles in the deep Earth[10].

Our study demonstrates that under extreme pressures of the deep mantle, the structure type becomes a main controlling factor of valence states, and the conventional sense of valence of elements could be altered to fit the structure. The new type of structure has the unexpected valence state of oxygen that is different from the O$^{2-}$ state in primary mantle oxides and silicates, and the ferrous, spin-paired, and non-magnetic state of iron that would affect Mg–Fe partitioning between mantle phases[46,47]. We may expect the new iron superoxide in pyrite structure to accommodate and form solid solutions with other divalent and trivalent cations such as Mg$^{2+}$ and Al$^{3+}$ (ref. [48]) and with anions, such as sulfur and halogens. We may further expect that the $Pa\bar{3}$ peroxide structure is only one example of the possible compositions in the deep Earth, and that additional structure types with different valence and spin, such as the recently discovered hexagonal phase[48], will emerge.

## Methods

**Sample synthesis and characterization.** The five samples of the pyrite-type phase have been synthesized and further probed by X-ray spectroscopic techniques in this study (Supplementary Table 1). Iron superoxide samples of FeO$_2$ or FeO$_2$H$x$ were synthesized from goethite (FeOOH) or $^{57}$Fe-enriched hematite (Fe$_2$O$_3$) mixture with O$_2$ and water H$_2$O, respectively, at 1800–2200 K under target pressures in laser-heated DACs at 16IDB, 13ID-D and High Pressure Synergetic Consortium (HPSynC) of the Advanced Photon Source of Argonne National Laboratory. In order to avoid any signal contamination from other phases in the sample and to assure complete transition, the whole starting materials of goethite (FeOOH) or hematite (Fe$_2$O$_3$) have been laser-heated repeatedly for 2–3 h at target pressures to synthesize a pure pyrite-structured FeO$_2$H$x$ ($x = 0$–1) phase.

XRD patterns were collected with the mapping steps of 4 μm for the whole-synthesized samples at beamlines 13ID-D or 16ID-B of Advanced Photon Source (APS), Argonne National Laboratory (ANL), confirming that the whole sample is the pyrite-type phase without any detectable amount of starting materials goethite or hematite left in the DACs (Supplementary Figures 1–5). The hydrogen content ($x$) of the iron superoxide samples that were synthesized from FeOOH or Fe$_2$O$_3$

with $H_2O$ was estimated to be 0.5–0.8 based on the unit-cell parameter calibration[10] (Supplementary Table 1). In particular, a relatively small size of starting materials were selected to be 25–30 μm in diameter with a thickness of 6–8 μm, which can reduce the synthesis time. We note that the X-ray beam was focused down to 5–7 μm in the full-width at half-maximum (FWHM) at the sample position. Thus the sample center was deliberately probed in order to achieve a relatively high efficiency of the spectroscopic measurements. That is, about one fifth of the volume of the synthesized samples was probed in the spectroscopic measurements.

**Synchrotron X-ray diffraction experiments**. XRD experiments were carried out at beamlines 13-IDD and 16-IDB of the Advanced Phonon Source, Argonne National Laboratory. A highly monochromatized incident X-ray beam was used with an energy of 30.49 keV (0.3738 Å), 33.17 keV (0.3738 Å), or 37.08 keV (0.3344 Å). The incident X-ray beam reached the sample position with a beam size of 2–5 μm in FWHM. For laser-heating XRD, two infrared laser beams were focused down to ~15 μm and 25–30 μm in the flat top area of the heating profile on both sides of the sample at beamlines 13-IDD and 16-IDB, respectively. With the help of the X-ray induced luminescence on the sample and/or ruby, both laser beams were co-axially aligned with the incident X-ray beam. The temperature of the samples was calculated through fitting the measured thermal radiation spectra with the gray-body approximation[49,50]. The temperature uncertainty is 100–200 K. A tiny piece of gold was placed in the sample chamber and its lattice parameters of gold were applied to determine pressure and uncertainty[51].

**Synchrotron Mössbauer spectroscopy experiments**. The SMS spectra of $^{57}Fe$-enriched Py-phase and $Fe_2O_3$ samples were collected at beamline 16-IDD, APS, ANL. A monochromatic X-ray beam with an energy of 14.41 keV, a bandwidth of 2 meV, and 5–7 μm in FWHM was used to excite the $^{57}Fe$ nuclei in the sample[52]. An avalanche photodiode detector (APD) was used to collect the time-delayed SMS signals in the forward direction with a typical collection time of ~2–4 h for each pressure. Pressure before and after each measurement was determined based on the Raman spectra of the diamond anvils where the sample contacted[53]. Py-phase samples were synthesized from the assemblages of $Fe_2O_3$ and $H_2O$ or $O_2$ approximately at 90–100 GPa and 2000 K. After the SMS spectrum of the sample was collected, a thin stainless-steel foil or a platelet of sulfate heptahydrate ($FeSO_4 \cdot 7H_2O$) was placed on the downstream side of the DAC to serve as a reference for the isomer shift measurements. Mössbauer hyperfine parameters, including quadrupole splitting, isomer shift, and magnetic hyperfine field of the samples were derived using the CONUSS program[52].

**X-ray absorption spectroscopy experiments**. Why use XAS measurements, collected in partial fluorescence yield mode (XAS-PFY), were carried out at beamline 16-IDD, APS, ANL. A monochromatic X-ray beam of at 7112 eV with 1 eV bandwidth from a water-cooled diamond (111) double-crystal mono-chromator was focused down to 5–7 μm in FWHM at the sample position. The XAS-PFY spectra were collected by setting the Bragg angle of the Si (333) analyzer to the maximum of the $K\alpha_1$ emission line while scanning the incident energy with an energy step of 0.4 eV through the Fe K pre-edge between 7086 and 7186 eV (Fig. 2a). The XAS-PFY spectra of Py-phase were collected at 133 GPa for the sample synthesized from $Fe_2O_3$ in excess $H_2O$ and 53–81 GPa for the sample synthesized from $Fe_2O_3$ in the pure $O_2$ medium.

**X-ray Raman scattering experiments**. XRS measurements on oxygen K-edge spectra of the Py-phase were conducted at 110 GPa by using the XRS technique at beamline 16ID-D, APS, ANL. With the new $320 \times 400$ mm IDT mirror, the inci-dent monochromatic X-ray beam were focused down to 5–7 μm in FWHM at the sample position. With the sample in a diamond-anvil cell mounted on the rotation center of the XRS spectrometer, oxygen K-edge spectra were collected by scanning the incident beam energy from 518 to 578 eV above the analyzer elastic scattering energy of 10.4467 keV, with an energy interval of 0.5 eV. The spectra were collected and averaged over 40 h of irradiation time. The Py-phase for the XRS measure-ments was synthesized from FeOOH (goethite) at 110 GP and 2000–2200 K using laser-heating systems at 13ID-D and HPSynC. LiF platelets were deliberately used as pressure-transmitting medium and thermal insulating layers. LiF does not contain any oxygen and would not contribute any signals to XRS measurements on oxygen K-edge spectra of the Py-phase.

**X-ray emission spectroscopy experiments**. High-pressure XES measurements of the Py-phase at 300 K were performed at beamline 16-IDD, APS, ANL using a Rowland circle configuration with a spherical silicon (4 4 0) analyzer crystal with resolution of ~1 eV. A helium tube was used to reduce scattering by air. An incident X-ray beam with an energy of 11.3 keV was used to excite the emission. The collection time for each XES spectrum was ~1 h. The 4–7 spectra were added for good statistics at a given pressure. Raman spectra of the diamond anvils were collected as a pressure gauge before and after each measurement while the ruby scale was cross-checked below 60 GPa. The Py-phase samples were synthesized from $Fe_2O_3$ and $H_2O$ approximately at 100–110 GPa and 2000 K. The XES

spectrum of the Py-phase at the highest pressure of 133 (±4) GPa was used as the reference for the low-spin state. The total spin moment was then evaluated using the integrated spectral area from 7030 eV to 7048 eV with respect to that of the low-spin reference (Supplementary Figure 9).

**Theoretical simulations**. Now with experimental knowledge of materials as the base, theory can successfully provide accurate and quantitative details in the extended pressure-temperature regime inaccessible to experiment[54]. The structure of $FeO_2$ has attracted many computational studies. However, an accurate description about its electron-correlation part in calculating the total energy still remains a challenging task. Based on previous studies, O 2p orbitals may hybrid with Fe 3d orbitals at the Fermi surface. It may cause O electrons to contribute to the electron-correlation portion in calculating the total energy. We note the linear-response method is excellent in dealing with the exchange-correlation energy for one atom species (e.g. Fe), although it may encounter issues for two types of atoms at the same time. Therefore, the use of Hubbard parameter U gives a first-order approximation to this energy portion. Combined with the self-consistent linear-response approach[55,56], many experimental observations can be reproduced by theory.

First-principles calculations based on density function theory using the package VASP were carried out using the projector-augmented wave pseudopotential and a plane-wave cutoff energy of 600 eV. The exchange-correlation functional employs the parameterization by Perdew, Burke, and Ernzerhof under the Generalized Gradient Approximation. We follow our previous computational recipe[18] to approximate the on-site Coulomb interaction by the Hubbard method ($U = 5.0$ and $J = 0.8$ eV). In parallel, we also employed the Heyd-Scuseria-Ernzerhof screened hybrid functional (HSE) that was recently adopted for a variety of iron-oxides and calculated reliable structural, magnetic, and electronic properties[57]. For HSE-type functional, the screening variable was set to 0.15 since it produces reasonable values in other Fe–O minerals like wüstite, magnetite, hematite, and goethite. We generated k-point in gamma centered Monkhorst-Pack grids of $7 \times 7 \times 7$ throughout our simulation. The structures were relaxed for atomic position, cell shape, and volumes for target pressures until the force acting on each atom was smaller than 0.01 eV/Å. Following the Bader charge division scheme, we are able to determine the valence state of Fe and O atoms from the electron charge density (Table S1). Calculated valence states definitely are against ferric ($Fe^{3+}$) or ferryl ($Fe^{4+}$) state of iron. Iron and oxygen atoms in pyrite-structured $FeO_2$ exhibit partial covalent feature, which is recognized as a common feature in pyrite $FeS_2$ model[37]. Therefore, our simulation results confirm that the valence state of iron in the Py-phase is close to $Fe^{2+}$ with O–O bonding.

## Data availability
The datasets generated during and/or analysed during the current study are available from the corresponding authors and J.L. (Jin.Liu@hpstar.ac.cn).

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

## Acknowledgements

We thank A. Shahar for providing the $^{57}Fe$-enriched hematite ($Fe_2O_3$ with $^{57}Fe$ of >96.5%) powder and X. Wu for $CaFeO_3$ samples. We acknowledge E.E. Alp, C. Kenney-Benson, T. Gu, and E. Greenberg for experimental assistance and helpful discussions. SMS, XAS, XES, and XRD measurements were performed at the High Pressure Collaborative Access Team (HPCAT 16-IDB and 16-IDD), APS, ANL. HPCAT operations are supported by DOE-NNSA's Office of Experimental Sciences. Y.X., P.C., and Y.M. acknowledge the support of DOE-BES/DMSE under Award DE-FG02-99ER45775. A portion of the XRD experiments was performed at GeoSoilEnviroCARS (Sector 13ID-D) at the APS, ANL. GeoSoilEnviroCARS is supported by the National Science Foundation —Earth Sciences (EAR-1128799) and Department of Energy—Geosciences (DE-FG02-94ER14466). The Advanced Photon Source is a U.S. Department of Energy (DOE) Office of Science User Facility operated for the DOE Office of Science by Argonne National Laboratory under Contract No. DE-AC02-06CH11357. The computational work was conducted on the SR10000-K1/52 supercomputing facilities of the Institute for Materials Research, Tohoku University. W.L.M. and J.L. acknowledge support from the NSF Geophysics Program (EAR 1446969) and the Deep Carbon Observatory. H.K.M. and Q.H. were supported by NSF Grants EAR-1345112, EAR-1722515, and EAR-1447438. This work was also partially supported by the National Natural Science Foundation of China (Grant no. U1530402).

## Author contributions

J.L., Q.H., L.Y., Y.X., P.C., Y.M., and V.B.P. carried out the experiment. J.L., W.B., Q.H., W.L.M., and H.-K.M. performed the experimental data analysis and interpretation. Q.H. performed the theoretical simulation. J.L., W.L.M., and H.-K.M. conceived and designed the project. J.L., Q.H., W.L.M., and H.-K.M. wrote the manuscript. All authors contributed to the discussion of the results and revision of the manuscript.

## Additional information

**Competing interests:** The authors declare no competing interests.

