## [Peer Review File · Nature Communications]

Reviewers' comments:

Reviewer #1 (Remarks to the Author):

This paper reports spectroscopic measurements carried out on an iron oxide (FeO₂) and a hydroxide (FeOOH_x) which adopts a similar crystallographic structure; both are stable only under high pressure (they are synthesized around 80 GPa and decompose below ~40 GPa). It follows a series of papers published in the last two years by the same group of authors: ref 2, first synthesis of FeO₂ (Nature); ref 4, synthesis and estimate of elastic properties of FeOOH_x (Nature); ref 1, stability of FeOOH_x (PNAS); ref 18, ab initio-based study of FeO₂ (PRB). The questions I have tried to answer are the following: (i) is the paper providing new information on FeO₂ and FeOOH_x? (ii) is this information accurate? (iii) are these properties important for the Earth (which justifies a publication in a broad interest journal such as Nat. Comm)?

(i) Yes, new experimental data on FeO₂ and FeOOH_x are reported in this paper. Synchrotron Moessbauer measurement show that FeOOH_x is non-magnetic. X-ray absorption spectroscopy at the Fe K-edge has been carried out on FeOOH_x and FeO₂: the XANES gives information on the valence of Fe in these compounds. X-ray emission spectroscopy suggests a low-spin state for Fe in FeOOH_x, in line with the Moessbauer measurements. Synchrotron Raman spectroscopy suggests the existence of an O-O dimer in FeOOH_x, confirming the previous X-ray diffraction measurements.

(ii) No, I don't find the information very accurate. This is not the author's fault: it is hard to determine the valence of elements with in situ techniques available for diamond anvil cell. But the limitations of the techniques should be discussed, along with more quantitative analysis. The reported data are mostly interpreted on the basis of qualitative comparison with standards under ambient conditions. I am not an expert in Synchrotron Moessbauer Spectroscopy (different from conventional Mossbauer spectro.): it is not clear to me why the fact that measured QS and IS factors are close to pyrite-FeS₂ implies that Fe electrons are in a low spin configuration. Values of these parameters may be more scattered than what is plotted on Fig. 1 (see Lin et al. Am Min 2009, which reports different values for QS from what is plotted). I am more familiar with XAS: the XANES is sensitive to valence, but also to other factors (spin, structure, volume) and a straightforward comparison with some standards at 0 GPa might be misleading (see for instance PRB 94, 014112). BTW I believe that it is the epsilon-Fe XANES which is plotted in Fig. 2A, and not the alpha-Fe XANES. No XRD spectrum is shown, even if XRD measurements are quantitatively discussed in the text (FWHM l. 137, O-O distance l. 150-160). XRD data MUST be shown for all samples studied. It is important also because in Refs 1-4, superoxides are observed mixed with reactants after laser-

heated diamond anvil cell synthesis (eg FeO₂ is not pure). Is it also the case here? Are other phases in the sample contaminating spectroscopic measurements? The ab initio study does not bring any new insight: Ref 18 (same group) was more complete and accurate. It was shown, in particular, that the predicted electronic properties for FeO₂ are tuned by the value of U parameter used in the DFT+U calculations.

(iii) In my view, the magnetic behavior, or valence of iron in FeO₂ superoxide would have a very limited influence on the thermodynamics or mechanics properties of the lowermost mantle, in case it forms (which would be in minor amounts – it is formed in the Fe-O system in excess oxygen, which is not the case in the mantle's volume). The fact that FeO₂ likely holds Fe(2+) ions and O₂(2-) dimers was already predicted in refs 2 and 18; having experimental data strengthening this description is interesting but not revolutionary. It is not the first example where iron valence does not follow oxygen fugacity (as mentioned in the paper, such an apparent paradox exists for Bridgmanite, ref 5). It is a new chemistry from a textbook point of view, but not for researchers in high pressure mineralogy and chemistry.

To sum up, this paper presents new experimental data on interesting compounds. They are in line with their recent description by the same group, and complement them, and with some ab initio predictions (but not all, which would have been interesting to discuss). A more detailed data discussion would have been useful; more data should be presented to strengthen this analysis (in particular, XRD). This can be made in an extended article in a more specialized journal.

I have found several sentences which, in my view, are misleading or even inexact:

- L. 44 “the spin-pairing transition drastically impacts the geochemical, geophysical and geodynamic process”: this is not accepted by the mineral physics community (see Liu Nat Comm 9, 1284, 2018).
- L. 50 “massive oxygen reservoirs”: what do the authors mean by “massive”: the reservoir is formed by oxygen carried down the mantle by subducted slabs (numbers?)?
- L. 61 “ab initio calculations without experimental constraints are unlikely to give the correct answers for the particularly difficult problems of iron oxides as demonstrated by the well-known example of FeO which is predicted to be a metal by all ab initio calculations”. This is obviously

incorrect (see Cocchioni PRB 71 2005). The authors themselves use DFT techniques which prevent this error in their own calculations (namely DFT+U (for strongly correlated systems) and hybrid HSE (for band-gap problem)).

- L. 148 Bader charge analysis underestimates the charge transfer (as in the case of alkali halides, see Henkelmann, Comput. Mater. Sci. 2006, 36, 354), and can therefore be misleading

- L. 162 I don't understand the sentence "forming the previous called 6+2 type bonding as shown in fig 3A"

- L. 184 In ref 29 I have found values of O-O distance up to 1.5 angstrom for bonded O-O dimer, in CdO₂: 1.4 should be replaced with 1.5

- L. 215 How does Fe valence in FeO₂ "change the meaning of siderophile"? please develop.

- The first article which predicts the stability of FeO₂ under pressure in the Fe-O₂ system is J. Phys.: Condens. Matter 27 (2015) 455501, which should be cited.

- Please correct "valance" where needed

Reviewer #2 (Remarks to the Author):

The manuscript by Liu et al. reports valuable data on new phases, FeO₂ and FeO₂H_x, collected at high pressure. Advanced spectroscopic techniques allowed the authors to probe critical aspects of the recently discovered material, including charge distribution, bonding, oxidation and spin state,

magnetism. Computational results are also provided in support of the proposed bonding and charge distribution.

The data presented is rather exceptional allowing to draw a comprehensive picture of the bonding and electronic state of the material. The manuscript however is substandard with respect to the quality of writing and discussion, making it difficult to properly evaluate its scientific content. Therefore it is my opinion that while data are interesting, the manuscript is not ready for publication. The review system should not be used to revise rough drafts. The authors should thoroughly revise this draft and submit it for a fresh review.

About the assessment of the samples composition, it is not clear if all samples were probed by XRD before conducting the spectroscopic measurements and if the authors made sure that the sample volume probed by XRD and by other techniques was the same. Please report in the figures or in a table in supplementary material the H content of the sample each set of measurement was performed on.

Please revise the abstract, it is very poorly written. It is not quite true that oxygen is constant at 2- in all other oxides.

36: revise

43: I don't know what Pauli's exclusive energy is. Are you suggesting overcoming Pauli's Exclusion rule?!

71: I do not understand this sentence

72-for 79 This very generic introduction to the techniques is not appropriate in this paper.

99: typo

102: revise

105: bridging ligation?

111: revise

114: samples with different H contents were probed at very different pressures. While this does not affect the overall picture, you should comment on what the effect of pressure might be based on the large body of literature on the pressure-dependence of the valence states.

121: why would oxygen contaminate the samples? are the other samples contaminated? perhaps you are concerned about probing py phase and free oxygen at the same time

130: because you did fit the spectra below decomposition you might have a good guess on the decomposition reaction. Besides, what is the evidence that a decomposition rather than a phase transition occurs? Was XRD performed on such material.

153: the equation is redundant

154: the O-O interatomic distance depends on both γ and a , why not considering the lattice parameter in your discussion? When discussing interatomic distances you should take into account pressure.

159: improved accuracy? show these better data in SI. The new D_0 value is by no means similar to the old one. Can the difference be explained by pressure?

161: I do not understand this sentence

163: Where is this conclusion coming from? Please add references if appropriate

183: there is a contradiction within this sentence, furthermore non interactive O-O distances have a lower limit not a well defined range

215: Changes the meaning of siderophile?? Please elaborate.

407: you cross-checked the ruby scale? maybe pressure obtained from the ruby scale

Reviewer #3 (Remarks to the Author):

The present study addresses the issue of changing chemistry of O, Fe, and H upon elevated pressures mimicking the conditions typical occurring in the middle Earth. The work was stimulated by a recent discovery of iron superoxide (FeO_2H_x with x from 0 to 1), considered as the major mineral in the deep Earth. In this manuscript, answers to several questions, which remained open so far, were presented including the valence states of Fe and O, nature of their chemical bonding, effect of hydrogen on the valence and bonding of Fe and O, the potential metallic character of superoxide, and possible spin pairing of d-electrons in superoxide. Employing synchrotron-based experimental techniques involving Mössbauer spectroscopy and pre-K-edge spectroscopy, it was found that the Fe is in a divalent form with d-electron spin-paired. On the other hand, the valence state of oxygen varied around -1. In addition, hydrogen is weakly bound in the structure, not affecting the valence state of iron. Other forms with peculiar valence and spin states are believed to exist due to various structures that become stable under such extreme conditions.

The work is well written and organized. The introduction covers the current knowledge and all the aspects regarding the existence of peculiar phases existing under high pressures (and/or temperatures). The experimental data are thoroughly interpreted and their results are complemented with theoretical predictions matching the conclusions drawn from the experiment.

Despite the well conducted study, there are two issues that remain open and could be addressed with data presented:

1). Below 40-50 GPa, the profile of the synchrotron Mössbauer spectra for $\text{FeO}_{2\text{Hx}}$ dramatically changes demonstrating emergence of the magnetic state (with typically repeating beats). The high-spin state is then stabilized. Is this transition accompanied by a change in the valence state of iron and/or oxygen? Is the octahedral coordination changing in terms of distortion? Is there any explanation why the change in the spin state is not sharp and shows the “pressure hysteresis”? What are the value of the Mössbauer hyperfine parameters of this (these) magnetic phase(s)? Are these values similar to those of other known iron oxide forms?

2). Can you speculate on the nature of these magnetic phases formed upon decrease in the pressure? Are these forms common in the Earth’s crust once we go to the Earth’s surface such as oxyhydroxides?

3). Can you guess the energy difference between t_{2g} and e_g orbitals stabilizing thus the high-spin state of Fe(II) ? Under such extreme pressures, are these orbitals still degenerate? What are the predictions from the theory?

Thus, a revision is required at this state. Once revised, it can be published in Nature Communications.

Reviewers' comments:

Reviewer #1 (Remarks to the Author):

This paper reports spectroscopic measurements carried out on an iron oxide (FeO₂) and a hydroxide (FeOOH_x) which adopts a similar crystallographic structure; both are stable only under high pressure (they are synthesized around 80 GPa and decompose below ~40 GPa). It follows a series of papers published in the last two years by the same group of authors: ref 2, first synthesis of FeO₂ (Nature); ref 4, synthesis and estimate of elastic properties of FeOOH_x (Nature); ref 1, stability of FeOOH_x (PNAS); ref 18, ab initio-based study of FeO₂ (PRB). The questions I have tried to answer are the following: (i) is the paper providing new information on FeO₂ and FeOOH_x? (ii) is this information accurate? (iii) are these properties important for the Earth (which justifies a publication in a broad interest journal such as Nat. Comm)?

[Authors]: We greatly appreciate that the reviewer recognizes the progress on pyrite-structured FeO₂H_x (x = 0 to 1) by our group. This newly discovered phase that forms under deep-mantle conditions advances our understanding about the deep Earth. **These high-pressure phases were independently reproduced, and their significance recognized, by other world-leading deep Earth research groups: ref 11 (Nature); ref 14 (GRL). Their very unusual stoichiometry clearly reflects altered crystal chemistry, but some of the most basic chemistry, such as the valence and spin states of Fe and O, magnetic properties and the role of hydrogen in the lattice remain unknown.** For instance, the valence state of iron in pure FeO₂ has been highly debated over 2+ and 3+ by the two theoretical studies (Jang *et al.*, 2017; Streltsov *et al.*, 2017). Here we employed the state-of-the-art synchrotron x-ray spectroscopic techniques to provide new insight into these properties and provide valuable implications on the mineral modelling of Earth's interiors. Please find more detailed responses to the following comments.

(i) Yes, new experimental data on FeO₂ and FeOOH_x are reported in this paper. Synchrotron Moessbauer measurement show that FeOOH_x is non-magnetic. X-ray absorption spectroscopy at the Fe K-edge has been carried out on FeOOH_x and FeO₂: the XANES gives information on the valence of Fe in these compounds. X-ray emission spectroscopy suggests a low-spin state for Fe in FeOOH_x, in line with the Moessbauer measurements. Synchrotron Raman spectroscopy suggests the existence of an O-O dimer in FeOOH_x, confirming the previous X-ray diffraction measurements.

[Authors]: We agree with the reviewer's comments, but note that the previous x-ray diffraction (XRD) measurements are not sensitive to the existence of an O-O dimer. We appreciate the reviewer's recognition of our synchrotron x-ray Raman spectroscopy study on oxygen K-edge, which is the first experimental evidence for oxygen-oxygen interaction in FeO₂H_x.

(ii) No, I don't find the information very accurate. This is not the author's fault: it is hard to determine the valence of elements with in situ techniques available for diamond anvil cell. But the limitations of the techniques should be discussed, along with more quantitative analysis. The

reported data are mostly interpreted on the basis of qualitative comparison with standards under ambient conditions. I am not an expert in Synchrotron Moessbauer Spectroscopy (different from conventional Mossbauer spectro.): it is not clear to my why the fact that measured QS and IS factors are close to pyrite-FeS₂ implies that Fe electrons are in a low spin configuration. Values of these parameters may be more scattered than what is plotted on Fig. 1 (see Lin et al. Am Min 2009, which reports different values for QS from what is plotted). I am more familiar with XAS: the XANES is sensitive to valence, but also to other factors (spin, structure, volume) and a straightforward comparison with some standards at 0 GPa might be misleading (see for instance PRB 94, 014112). BTW I believe that it is the epsilon-Fe XANES which is plotted in Fig. 2A, and not the alpha-Fe XANES. No XRD spectrum is shown, even if XRD measurements are quantitatively discussed in the text (FWHM l. 137, O-O distance l. 150-160). XRD data MUST be shown for all samples studied. It is important also because in Refs 1-4, superoxides are observed mixed with reactants after laser-heated diamond anvil cell synthesis (eg FeO₂ is not pure). Is it also the case here? Are other phases in the sample contaminating spectroscopic measurements? The ab initio study does not bring any new insight: Ref 18 (same group) was more complete and accurate. It was shown, in particular, that the predicted electronic properties for FeO₂ are tuned by the value of U parameter used in the DFT+U calculations.

[Authors]: Many thanks for the reviewer's comments and suggestions. As the reviewer pointed out, to determine the valence states, bondings, and magnetic states, for Fe, O, H in samples under high pressures is a very challenging task. **Therefore, we used multiple complementally techniques to address each question. For instance, the QS (Fig. 2B) and IS (Fig. 2C) are not sufficiently strong indicators of spin state, but the clean, single beat SMS Mössbauer spectra (Fig. 2A) clear indicate the lack of magnetic ordering. The additional X-ray emission study (Fig. 3) clinches the low-spin diagnosis.**

Even at ambient conditions, the valence state are not easily determined and largely based on empirical experimental observations. In this work, we used the most advanced techniques that have been extensively used to investigate the valence state under high pressures. The valence state and magnetic behavior have been highly debated by theoretical studies (Jang et al. 2017 and Streltsov et al. 2017). With the help of a battery of state of art synchrotron x-ray spectroscopic techniques, we are dedicated to make a comprehensive understanding of these properties for the Py-phase.

Please find the modifications we have made accordingly as follows:

(1) We have added more discussion on the challenges and limitations of the spectroscopic techniques in this study along with more quantitative analysis. For example, please refer to the new "Methods" section and Lines 81-83, 90-94, 104-106, 121-124, 128-134 and 140-145.

(2) We did not intend to imply that Fe electrons are in a low spin configuration due to the measured QS and IS factors of FeO₂H_x close to pyrite-FeS₂. To clarify, we have removed the original phrase, "leading to a low-spin, nonmagnetic configuration".

(3) As suggested, the QS values for $(\text{Mg}_{0.75}, \text{Fe}_{0.25})\text{O}$ at 0-40 GPa from Lin et al., *Am. Miner.*, 2009, have been plotted in Figure 2. We note that it is in the low spin state for iron in $(\text{Mg}_{0.52}, \text{Fe}_{0.48})\text{O}$ by Solomatova *et al.* (2016), which are found to be close to that for FeO_2H_x .

(4) Sanson et al., *Phys. Rev. B* 94, 014112, 2016 reported that the pre-edge peak position of Fe_2O_3 hematite remains constant approximately at 7115.2 eV from 2.8 GPa and ~50 GPa in the high-spin state, while it keeps constant at 7114.5-7114.6 eV between 55 and 80 GPa in the low-spin state. It indicates a small decreases by 0.6-0.7 eV across the spin transition of iron in Fe_2O_3 as well as the pressure has a minimal effects on the pre-edge peak. Our measurements on the pyrite-type phase between 53 and 133 GPa exhibit the same trend with pressure. Therefore, it should be justified for the first order comparison to use some standards at 0 GPa. The XANES measurements of metallic iron were conducted at ambient conditions and it is in the alpha phase.

(5) XRD patterns for all samples in this study have been added (please see Supplementary Figures 1-5 and Supplementary Table 1). In order to avoid any signal contamination from other phases in the sample, the entire goethite (FeOOH) or hematite (Fe_2O_3) starting materials have been laser heated for 2-3 hours at target pressures to synthesize a pure pyrite-structured FeO_2H_x ($x = 0 - 1$) phase. In addition, the pressure medium was carefully selected. For instance, LiF was used as the pressure medium for XRS measurements on the oxygen *K*-edge of pyrite-structured FeO_2H_x . To clarify, the new subsection “*Sample synthesis and characterization*” has been added in Lines 231-252.

(6) Regarding the comments of “The ab initio study does not bring any new insight...,” we agree with the reviewer that Hubbard *U* plays an important role in tuning the calculated electronic structures. As Ref. 18 (Jang et al., 2017) stated out, FeO_2 changes from a metal to an insulator by increasing the *U* value. Our current study not only employed the previously used GGA+*U* ($U=5.0$ eV, $J=0.8$ eV) method, but also hybrid HSE functionals ($a = 0.15$) (Meng, 2016) to reach agreement in structural properties with experimental results, especially in describing the O-O bonding. However, it is true that at the current stage, we are not able to provide more insights on the electronic or magnetic structures of FeO_2 . The atomic structure and magnetic configuration from experiment were used the inputs for our simulation. Since the electronic structure is still under debate (Jang et al., 2017; Streltsov et al., 2017) and the Py-phase was wrongly calculated as a ferromagnetic phase (Nishi et al., 2017), we have to be very careful about the literature information. In this study, our theoretical simulations aim to confirm the O-O bonding in FeO_2H_x .

(iii) In my view, the magnetic behavior, or valence of iron in FeO_2 superoxide would have a very limited influence on the thermodynamics or mechanics properties of the lowermost mantle, in case it forms (which would be in minor amounts – it is formed in the Fe-O system in excess oxygen, which is not the case in the mantle’s volume). The fact that FeO_2 likely holds $\text{Fe}(2+)$ ions and $\text{O}_2(2-)$ dimers was already predicted in refs 2 and 18; having experimental data strengthening this description is interesting but not revolutionary. It is not the first example where iron valence does not follow oxygen fugacity (as mentioned in the paper, such an apparent

paradox exists for Bridgmanite, ref 5). It is a new chemistry from a textbook point of view, but not for researchers in high pressure mineralogy and chemistry.

[Authors]: As the reviewer pointed out, FeO₂H_x represents interesting new chemistry. We would also like to contend that it is not just textbook knowledge but is also potentially important to the deep Earth chemistry. However, its nature and crystal chemistry have been largely unknown or highly debated.

Regarding theoretical predications on the valence state and magnetic behavior of iron in pure FeO₂, the original ref. 2 (Hu et al., *Nature*, 2017) did not include such information and the original ref. 18 (Jang et al., *PRB*, 2017) suggested the iron valence to be 2+ based on the analogy to pyrite FeS₂. Later on, theoretical calculations by Streltsov et al., *Sci. Rep.*, 2017 argued that iron in Fe₂O is ferric iron (Fe³⁺), making it much more magnetic and qualitatively distinct from its analogue - pyrite FeS₂.

Here we first present experimental evidence to clarify the discrepancy between the two theoretical studies. More importantly, the two theoretical studies did not cover the effects of hydrogen on the valence state of iron in hydrogen-bearing FeO₂H_x. Experiments in this study for the first time observe that the incorporation of hydrogen in FeO₂ has minimal effects on the valence state of iron. It is also worth noting that iron valence does not follow oxygen fugacity for FeO₂H_x due to oxygen-oxygen interactions in the lattice, which is completely new and different from the mechanism for bridgmanite incorporating ferric iron Fe³⁺.

We agree with the reviewer on that the magnetic behavior, or valence of iron in FeO₂H_x superoxide may be less linked to the thermodynamic or mechanical properties of the lowermost mantle. However, those properties are important considerations with studying the stability of FeO₂H_x at high pressure. Furthermore, knowledge of the physical and chemical properties of mantle phases would advance our understanding of the dynamics, composition, structure, and evolution of our planet. In particular, the presence of FeO₂H_x and the distinct interaction of hydrogen in its lattice may play a key role in the global circulation of water (and hydrogen and oxygen). The distinct chemical state of FeO₂H_x would also make Earth's deep mantle chemistry more complex.

It is well established that FeO₂H_x would form through goethite transformation and high-pressure reaction between water and iron (or a series of iron oxides from FeO to Fe₂O₃) under deep-mantle conditions (Hu *et al.*, 2016; Nishi *et al.*, 2017; Mao *et al.*, 2017; Yuan *et al.*, 2018). What's more, our group recently found that ferropericlase, which is the second most abundant mantle phase, reacts with water (from hydrous materials) to form pyrite-type phases in the deep mantle (these new results will be reported in another manuscript under review). Therefore, the amount of FeO₂H_x appears largely limited to the abundance and distribution of water in the lower mantle. Thus far, the water inventory in the deep mantle has not been well constrained. It is possible that FeO₂H_x could represent an important endmember in 'wetter' portions of the deep mantle. We note that FeO₂H_x could be an important phase for controlling the transport, storage, and release of volatiles even in regions where it is a minor phase.

To sum up, this paper presents new experimental data on interesting compounds. They are in line with their recent description by the same group, and complement them, and with some ab initio predictions (but not all, which would have been interesting to discuss). A more detailed data discussion would have been useful; more data should be presented to strengthen this analysis (in particular, XRD). This can be made in an extended article in a more specialized journal.

[Authors]: Based on comments and suggestions by the three reviewers, XRD patterns for all samples probed in this study and more discussion on data have been added.

I have found several sentences which, in my view, are misleading or even inexact:

- L. 44 *“the spin-pairing transition drastically impacts the geochemical, geophysical and geodynamic process”*: this is not accepted by the mineral physics community (see Liu Nat Comm 9, 1284, 2018).

[Authors]: Liu et al. (2018) observed abrupt changes on density, bulk sound velocity, and electrical conductivity across the spin transition of iron in a ferric-iron-only bridgmanite at 300 K. Their study further suggested those changes would be softened due to the temperature-induced broadening of the spin crossover, which is well known among the mineral physics community. On the other hand, there are more recent studies that found the spin transition of iron can largely affect the Mg-Fe partitioning, melting, and viscosity behaviors of mantle phases (e.g., Lin et al., *Rev. Geophys.* 51, 244-275, 2013; Irifune et al., *Science* 327, 193-195, 2010; Nomura et al., *Nature* 473, 199-202, 2011; Wu and Wentzcovitch, *PNAS* 111, 10468-10472, 2014; Deng and Lee, *Nature Commun.* 8, 1997, 2017).

To be consistent with the literature, the original sentence has been revised to, “which can affect the physical, chemical, and transport properties of mantle phases.”

- L. 50 *“massive oxygen reservoirs”*: what do the authors mean by “massive”: the reservoir is formed by oxygen carried down the mantle by subducted slabs (numbers?)?

[Authors]: We realized that the description of “massive” is too vague. It has been removed for the avoidance of confusion. To clarify, “oxygen reservoirs” has been changed to “oxygen-rich reservoirs”. Oxygen-rich reservoirs may develop in the deep mantle through the formation of the oxygen-rich phase of pyrite-type FeO_2H_x in the presence of water (or hydrous materials), instead of free oxygen. The water is likely transported in descending subducted plates to the base of the mantle in the form of aluminous hydrous silicates (for example, phase H) (Nishi et al., *Nature Geo.* 7, 224-227, 2014; Walter et al., *Chem. Geol.* 418, 16-29, 2015).

- L. 61 *“ab initio calculations without experimental constraints are unlikely to give the correct answers for the particularly difficult problems of iron oxides as demonstrated by the well-known example of FeO which is predicted to be a metal by all ab initio calculations”*. This is obviously incorrect (see Coccochioni PRB 71 2005). The authors themselves use DFT techniques which

prevent this error in their own calculations (namely DFT+U (for strongly correlated systems) and hybrid HSE (for band-gap problem)).

[Authors]: We appreciate the reviewer pointing out the inaccuracy for the description of the computational complexity of the FeO₂ system. In the revised manuscript, the original sentence has been removed while more details are added. Please refer to Lines 60-66 and 314-324.

- L. 148 Bader charge analysis underestimates the charge transfer (as in the case of alkali halides, see Henkelmann, Comput. Mater. Sci. 2006, 36, 354), and can therefore be misleading

[Authors]: We agree with the reviewer that even in the case of well-known FeS₂, charges of anions are underestimated. In the revision, more discussion has been added to point out this underestimation and soften our tone in the description of O-O covalent bonding. Please refer to Lines 150-154.

- L. 162 I don't understand the sentence "forming the previous called 6+2 type bonding as shown in fig 3A"

[Authors]: The "6+2 type bonding" indicate that each Fe atom connects with 6 O atoms in the length of 1.79 Å and 2 further O atoms in the length of 2.69 Å (at 75 GPa) like in the pyrite-SiO₂. However, we shall note that the longer Fe-O distance of 2.69 Å in FeO₂ does not establish a valid bond. It is not comparable with pyrite-SiO₂. The original sentence has been revised to "With $0.35 < y < 0.39$, iron-oxygen octahedron is chained by the O-O bonds as shown in Fig. 3A."

- L. 184 In ref 29 I have found values of O-O distance up to 1.5 angstrom for bonded O-O dimer, in CdO2: 1.4 should be replaced with 1.5

[Authors]: "1.4" has been corrected to "1.53".

- L 215 How does Fe valence in FeO2 "change the meaning of siderophile"? please develop.

[Authors]: We intended to state the low-spin Fe in FeO₂ may influence the partitioning behavior of iron, as observed in other mantle phases due to the spin crossover of iron (e.g., Irifune et al., *Science*, 2010 and Nomura et al., *Nature*, 2011). The use of "siderophile" here is not straightforward and it has been changed into "the ferrous, spin-paired and non-magnetic state of iron that would affect Mg-Fe partitioning among mantle phases^{40,41}."

- The first article which predicts the stability of FeO2 under pressure in the Fe-O2 system is J. Phys.: Condens. Matter 27 (2015) 455501, which should be cited.

[Authors]: It has been credited and cited. Many thanks for pointing out the missing reference.

- Please correct "valance" where needed

[Authors]: Corrected.

Reviewer #2 (Remarks to the Author):

The manuscript by Liu et al. reports valuable data on new phases, FeO₂ and FeO₂H_x, collected at high pressure. Advanced spectroscopic techniques allowed the authors to probe critical aspects of the recently discovered material, including charge distribution, bonding, oxidation and spin state, magnetism. Computational results are also provided in support of the proposed bonding and charge distribution.

The data presented is rather exceptional allowing to draw a comprehensive picture of the bonding and electronic state of the material. The manuscript however is substandard with respect to the quality of writing and discussion, making it difficult to properly evaluate its scientific content. Therefore it is my opinion that while data are interesting, the manuscript is not ready for publication. The review system should not be used to revise rough drafts. The authors should thoroughly revise this draft and submit it for a fresh review.

[Authors]: Many thanks for the reviewer's positive feedback on our study and advice for a thorough revision. The manuscript has been revised based all the reviewers' comments and suggestions as well as editorial guidelines.

About the assessment of the samples composition, it is not clear if all samples were probed by XRD before conducting the spectroscopic measurements and if the authors made sure that the sample volume probed by XRD and by other techniques was the same. Please report in the figures or in a table in supplementary material the H content of the sample each set of measurement was performed on.

[Authors]: Before conducting the spectroscopic measurements, the whole samples have been scanned by XRD to confirm that all the starting materials have been transformed to the pyrite-type phase. A prolonged laser heating period has been applied, in general, 2-3 hours for each sample synthesis. In particular, the sample center was deliberately probed in order to achieve a relatively high efficiency of the spectroscopic measurements. To clarify, the new section of "Sample synthesis and characterization" has been added in the main text. Furthermore, in Supplementary Materials have been provided the representative XRD patterns (Supplementary Figures 1-5) and new Supplementary Table 1 for sample synthesis conditions and H content.

Please revise the abstract, it is very poorly written. It is not quite true that oxygen is constant at 2- in all other oxides.

[Authors]: The abstract has been thoroughly rewritten. In particular, "at 2- like other oxides" has been changed into to "at 2- like mantle ferroperricite and silicates".

36: revise

[Authors]: It has been revised to "It is conventionally accepted that oxygen anion has an invariable 2- valence state in mantle ferroperricite and silicates..."

43: *I don't know what Pauli's exclusive energy is. Are you suggesting overcoming Pauli's Exclusion rule?!*

[Authors]: We apologize for the inaccurate description and the phrase of “*could overcome the Pauli's exclusive energy*” has been removed.

71: *I do not understand this sentence*

[Authors]: The original sentence describes how the incident x-ray beam reaches the sample in a diamond anvil cell. It may be redundant and has been removed together with what were Lines 72-79 in original manuscript.

72-for 79 *This very generic introduction to the techniques is not appropriate in this paper.*

[Authors]: This text has been removed.

99: *typo*

[Authors]: It has been corrected to “valence”.

102: *revise*

[Authors]: The sentence has been revised into two sentences (Lines 81-85).

105: *bridging ligation?*

[Authors]: It shall be “the ligand-ligand interaction”.

111: *revise*

[Authors]: This has been revised (Lines 90-94). In particular, the original phrase of “*consistent with the Fe²⁺ ferrous conclusion of the SMS measurements*” has been removed.

114: *samples with different H contents were probed at very different pressures. While this does not affect the overall picture, you should comment on what the effect of pressure might be based on the large body of literature on the pressure-dependence of the valence states.*

[Authors]: It has been compared with the two major mantle phases, ferropericlasite and bridgmanite. Please refer to Lines 90-94.

121: *why would oxygen contaminate the samples? are the other samples contaminated? perhaps you are concerned about probing py phase and free oxygen at the same time*

[Authors]: Oxygen is not going to contaminate the samples. Regarding “to avoid contamination” here, it would be a concern for the oxygen K-edge x-ray Raman scattering (XRS) signals if the pressure medium contained oxygen. Thus we deliberately used pressure medium LiF, instead of H₂O or O₂, together with our FeOOH sample in the sample chamber for XRS measurements at high pressures. To clarify, the original sentence has been revised (Lines 103-105).

130: because you did fit the spectra below decomposition you might have a good guess on the decomposition reaction. Besides, what is the evidence that a decomposition rather than a phase transition occurs? Was XRD performed on such material.

[Authors]: According to XRD patterns for FeO₂ collected upon decompression that are reported in another manuscript (under review), the diffraction peaks of the pyrite-type FeO₂ became broadened and diminished approximately from 59 GPa to 31 GPa, while some new peaks appeared below 41 GPa and can be indexed to the hematite (Fe₂O₃) structure. Thus FeO₂ likely undergoes decomposition, instead of a phase transition below ~40 GPa.

153: the equation is redundant

[Authors]: The equation clearly demonstrates the relation of D_o with y and a , and further with pressure through a . It may be redundant, but we feel it is beneficial to readers with little background in this relationship.

154: the O-O interatomic distance depends on both y and a , why not considering the lattice parameter in your discussion? When discussing interatomic distances you should take into account pressure.

[Authors]: We agree with the reviewer that pressure needs to be taken into account when comparing the O-O distances at high pressures. We follow previous strategies of using the oxygen coordination y , which is a function of O-O distance and lattice parameter a . We note that the a is a function of pressure. The use of y has considered pressure effects when including the a in the equation. The sentence has been rewritten to clarify the term (Lines 154-159):

“For pyrite-type structures, such covalent nature is mainly controlled by the anion-anion distance (D_o). We follow previous strategies and use the fractional coordination y of oxygen to compare O-O distance from difference pressures: ”

159: *improved accuracy? show these better data in SI. The new Do value is by no means similar to the old one. Can the difference be explained by pressure?*

[Authors]: A more pure FeO₂ sample was synthesized in this study and the new x-ray diffraction pattern of FeO₂ are of better quality (Supplementary Figure S2). However, as the reviewer pointed out, this does not necessarily means it is more accurate. We found it is consistent with our previous experimental value with a small difference of 3%, which is reasonable and likely due to pressure effects. The original sentence has been revised (Lines 163-166).

161: *I do not understand this sentence*

[Authors]: The “6+2 type bonding” indicate that each Fe atom connects with 6 O atoms in the length of 1.79 Å and 2 further O atoms in the length of 2.69 Å (at 75 GPa) like in the pyrite-SiO₂. However, we shall note that the longer Fe-O distance of 2.69 Å in FeO₂ does not establish a valid bond. It is not comparable with pyrite-SiO₂. The sentence original sentence has been revised to “With $0.35 < y < 0.39$, iron-oxygen octahedron is chained by the O-O bonds as shown in Fig. 3A.”

163: *Where is this conclusion coming from? Please add references if appropriate*

[Authors]: The position of hydrogen in the lattice is based on theoretical simulations by Zhu et al., JACS, 2017. The missing reference has been added here.

183: *there is a contradiction within this sentence, furthermore non interactive O-O distances have a lower limit not a well defined range*

[Authors]: Here we used the examples of well-defined peroxide and PdF₂-type dioxides to explain the O-O interaction. The original sentence has been revised (Lines 190-192).

215: *Changes the meaning of siderophile?? Please elaborate.*

[Authors]: We intended to state the low-spin Fe in FeO₂ may influence the partitioning behavior of iron, as observed in other mantle phases due to the spin crossover of iron (e.g., Irifune et al., Science, 2010 and Nomura et al., Nature, 2011). The use of “siderophile” here is not straightforward and it has been changed into “the ferrous, spin-paired and non-magnetic state of iron that would affect Mg-Fe partitioning among mantle phases^{40,41}.”

407: *you cross-checked the ruby scale? maybe pressure obtained from the ruby scale*

[Authors]: The ruby scale was used to cross check at pressures below 50 GPa, while above 50 GPa it was not used due the relatively weak ruby fluorescence. To be consistent in pressure scale, the Raman spectra of the diamond anvils were collected as a primary pressure gauge before and after each measurement, because most of spectroscopic measurements were carried out beyond 50 GPa.

Reviewer #3 (Remarks to the Author):

The present study addresses the issue of changing chemistry of O, Fe, and H upon elevated pressures mimicking the conditions typical occurring in the middle Earth. The work was stimulated by a recent discovery of iron superoxide (FeO_2H_x with x from 0 to 1), considered as the major mineral in the deep Earth. In this manuscript, answers to several questions, which remained open so far, were presented including the valence states of Fe and O, nature of their chemical bonding, effect of hydrogen on the valence and bonding of Fe and O, the potential metallic character of superoxide, and possible spin pairing of d-electrons in superoxide. Employing synchrotron-based experimental techniques involving Mössbauer spectroscopy and pre-K-edge spectroscopy, it was found that the Fe is in a divalent form with d-electron spin-paired. On the other hand, the valence state of oxygen varied around -1. In addition, hydrogen is weakly bound in the structure, not affecting the valence state of iron. Other forms with peculiar valence and spin states are believed to exist due to various structures that become stable under such extreme conditions.

The work is well written and organized. The introduction covers the current knowledge and all the aspects regarding the existence of peculiar phases existing under high pressures (and/or temperatures). The experimental data are thoroughly interpreted and their results are complemented with theoretical predictions matching the conclusions drawn from the experiment. Despite the well conducted study, there are two issues that remain open and could be addressed with data presented:

1). Below 40-50 GPa, the profile of the synchrotron Mössbauer spectra for FeO_2H_x dramatically changes demonstrating emergence of the magnetic state (with typically repeating beats). The high-spin state is then stabilized. Is this transition accompanied by a change in the valence state of iron and/or oxygen? Is the octahedral coordination changing in terms of distortion? Is there any explanation why the change in the spin state is not sharp and shows the “pressure hysteresis”? What are the value of the Mössbauer hyperfine parameters of this (these) magnetic phase(s)? Are these values similar to those of other known iron oxide forms?

[Authors]: Regarding the decomposition of FeO_2 below 40 GPa, it is beyond the scope for this study, and will be reported in another manuscript that is under review.

To our best knowledge, available XRD and XES experimental observations indicate that the pyrite-type FeO_2 is not stable in the high-spin state and would decompose into Fe_2O_3 . That is, the high-spin FeO_2 is unstable and decomposes into oxygen and hematite, associated with a change in the valence states of iron and oxygen.

Regarding the question “Is the octahedral coordination changing in terms of distortion,” the structural strain that may be altered by a distortion in the octahedral coordination structure could cause the spin transition of iron in FeO_2H_x upon decompression. In general, the octahedral coordination distortion may cause a reduction in the energy gap between t_{2g} and e_g orbitals. As

long as the energy gap becomes smaller than the mean spin pairing energy, the high-spin state will be the ground state. Furthermore, our theoretical simulations indicate that that octahedral coordination is not changing upon the spin state, although the FeO₂ lattice is slightly distorted along the (111) for the high-spin state. Once FeO₂ is in the high-spin state, it will further decompose as experimental observations.

In addition, the XES spectra were collected upon decompression from 133 GPa to 30 GPa. It is not uncommon to observe the “pressure hysteresis” in the spin transition with decreasing pressure, which has also been found in (Fe,Mg)CO₃ carbonate by Lavina et al., *GRL* 36, L23306, 2009.

Lastly, the Mössbauer hyperfine parameters of the magnetic component are close to that of hematite (Fe₂O₃). These experimental observations suggest the decomposition of the pyrite-type FeO₂ below 40 GPa.

Please find the relevant discussion added correspondingly in the main text (e.g., Lines 111-134).

2). *Can you speculate on the nature of these magnetic phases formed upon decrease in the pressure? Are these forms common in the Earth’s crust once we go to the Earth’s surface such as oxyhydroxides?*

[Authors]: XRD patterns of FeO₂ collected upon decompression suggest that FeO₂ could decompose below 40 GPa to form hematite that is a common mineral on Earth. These results have been reported in our another manuscript (under review),

3). *Can you guess the energy difference between t_{2g} and e_g orbitals stabilizing thus the high-spin state of Fe(II)? Under such extreme pressures, are these orbitals still degenerate? What are the predictions from the theory?*

[Authors]: The energy difference between t_{2g} and e_g orbitals can be analytically computed by the crystal field theory (Yoshiki Iwazaki, 2014 PhD thesis, the University of Tokyo, *First-principles theoretical study of defect induced phenomena in perovskite-type dielectric oxides*). Taking the Fe-O bond length as 1.76 Å at 100 GPa for FeO₂, this ideal octahedral FeO₆ model gives an energy difference of 3.9 eV.

A better approximation comes from using the density functional theory. Here the energy difference would be a function of k -vector in the Brillouin zone. According to the following band structures, the t_{2g} and e_g states of Fe d orbitals are no longer degenerated at high pressures. Fe $3d$ electrons would fill the t_{2g} orbitals first and achieve a paired nonmagnetic state for Fe (II) atoms. The energy difference between t_{2g} and e_g is smaller than the number predicted by the crystal field theory due to the hybridization of Fe $3d$ and O $2p$ orbitals (Revision Fig. 1). The breakdown of dependency is much clearer in the density of states (Revision Fig. 2).

Revision Fig. 1. Band structure for FeO₂ (left) and FeO₂H (right) at 100 GPa.

Revision Fig. 2. Electronic density of states for FeO₂ (left) and FeO₂H (right) at 100 GPa.

It is worth mentioning that the interpretation of the electronic structure is still under debate (Jang *et al.*, 2017; Streltsov *et al.*, 2017). However, it is generally accepted that e_g and t_{2g} orbitals are no longer degenerated but separated with each other and hybrid with O $2p$ orbitals for the pyrite phase.

Thus, a revision is required at this state. Once revised, it can be published in Nature Communications.

[Authors]: We greatly appreciate the reviewer's suggestions and comments aforementioned.

Reviewers' comments:

Reviewer #1 (Remarks to the Author):

The authors have taken into account many of the three referees suggestions on the form and the manuscript is now easier to read. The inclusion of X-ray diffraction data, showing nearly pure FeO₂H_x py phases, makes some of the author's points stronger. Some claims remain unjustified (see below).

The claim that FeO₂H_x crystal chemistry has strong bearing for the Earth's mantle has been moderated in the new version. The authors still mention that this phase may play a role in the transport and storage of H and O in the mantle and affect its redox properties, without further detail. This is highly speculative. The reference to the middle Earth in the title is, in my view, not justified. The last sentence in the abstract is obscure (BTW the second referee asked to completely re-write the abstract, it has been only slightly modified). L. 39-42 make an artificial link between oxygen fugacity in the mantle and the synthesis of iron oxides with various stoichiometries in the laboratory: it should be removed.

This work is a very good high pressure crystal chemistry study, which confirms (after prediction ref 17) an interesting but not revolutionary behavior: the formation of an (O₂)₂- dimer in a high pressure ionic compound. This is in line with recent synthesis of metal peroxides such as MgO₂ stable only under pressure. The paper would certainly deserve publication, after further improvement of the form and science, rather in a chemistry or condensed matter journal than in a multidisciplinary journal.

Data analysis and discussion is still unsatisfactory. Two examples:

(1) L. 137-143, the disappearance of a XES magnetic signal with increasing pressure is described as an indication of a high spin to low spin transition (please correct typo) in FeO₂H_x around 50 GPa. However, the data have been collected on pressure decrease and FeO₂H_x, stable around 100 GPa, decomposes in the same range into magnetic compounds as discussed l. 130-134. No high spin-low spin transition in FeO₂H_x is needed.

(2) L. 163-167 a value of O-O bond length of 1.874(1) is reported, based on XRD data analysis. It is presented as an important output of the study. It is IMPOSSIBLE to obtain such accuracy with a Rietvelt refinement of a high pressure XRD powder spectrum, especially for the weakest scatterer of

the compound. No Rietvelt fit and goodness parameters are provided. The authors should better discuss the length predicted ab initio, if it is not sensitive to calculations parameters.

My role should not be to correct such elementary mistakes.

The paper is not well organized: XRD data are discussed in a paragraph called "Theoretical simulations..."

Reviewer #3 (Remarks to the Author):

In the revised version of the manuscript, the authors addressed thoroughly all the comments and remarks raised during the first critical assessment of the manuscript. The Reviewer is fully satisfied with the current version of the manuscript and recommends its publication in the Nature Communications journal as it is. The work will surely attract the attention of the scientific community in the respective and related fields and will definitely stimulate other studies.

Point-by-point responses to reviewers' comments:

Reviewers' comments:

Reviewer #1 (Remarks to the Author):

The authors have taken into account many of the three referees' suggestions on the form and the manuscript is now easier to read. The inclusion of X-ray diffraction data, showing nearly pure FeO₂H_x py phases, makes some of the author's points stronger. Some claims remain unjustified (see below).

[Authors]: We greatly appreciate Rev#1's recognition of the improved quality in the previously revised manuscript.

The claim that FeO₂H_x crystal chemistry has strong bearing for the Earth's mantle has been moderated in the new version. The authors still mention that this phase may play a role in the transport and storage of H and O in the mantle and affect its redox properties, without further detail. This is highly speculative. The reference to the middle Earth in the title is, in my view, not justified. The last sentence in the abstract is obscure (BTW the second referee asked to completely re-write the abstract, it has been only slightly modified). L. 39-42 make an artificial link between oxygen fugacity in the mantle and the synthesis of iron oxides with various stoichiometries in the laboratory: it should be removed.

[Authors]: As this study is focused on the high-pressure crystal chemistry of FeO₂H_x, we have minimized the statements for the potential role of the Py-phase in the deep mantle with the following changes:

(1) In the abstract (**Lines 26-28**), we removed "and play a new role in influencing mantle redox and heterogeneity."

(2) At the end of the "Discussion" section (**Lines 234-235**), we removed "Iron and oxygen in the new phase will play a dominant role in influencing the redox state and interaction between the iron core and oxide mantle in the middle Earth."

The title has been revised to "Altered chemistry of O, Fe, and H under deep Earth conditions." We have taken "in the middle Earth" out of the title and main text.

The last sentence has been clarified and the abstract has been revised (**Lines 21-30**):

"Drastically altered chemistry was observed in a recently discovered iron superoxide (FeO₂H_x with $x = 0$ to 1) under deep Earth conditions, but its puzzling crystal chemistry is largely unknown at high pressures. Here we present evidence that despite the unprecedentedly high O/Fe ratio in FeO₂H_x, iron remains in the ferrous, spin-paired and non-magnetic state at 60-133 GPa, while the presence of hydrogen has minimal effects on the valence of iron. The reduced iron is accompanied by oxidized oxygen due to oxygen-oxygen interactions. The valence of oxygen is

not -2 as in all other mantle minerals, instead it varies around -1. This result indicates that like iron, oxygen may have multiple valence states in our planet's interior. Our study suggests a possible change in the chemical paradigm of how oxygen, iron, and hydrogen behave under deep Earth conditions."

The link between the series of iron oxides with oxygen fugacity has been removed. In **Lines 33-40**, the original sentences have been revised to "It is conventionally accepted that the oxygen anion has an unvarying -2 valence state in mantle ferroperricite and bridgmanite throughout the deep interior, where the oxygen fugacity decreases with increasing depth. The redox states are mostly controlled by the 3d transition element Fe which could vary among three valence states, metallic Fe⁰, ferric Fe³⁺ and ferrous Fe²⁺. Recently, a series of new iron oxides have been found with varying O/Fe stoichiometry ranging from the end-member Fe³⁺₂O₃ found on our planet's highly oxidized surface to the other end-member Fe²⁺O which should be stable at the highly reduced conditions in the deep lower mantle down to the core-mantle boundary, which include Fe₅O₇, Fe₄O₅, and Fe₅O₆ (refs 2-4)."

This work is a very good high pressure crystal chemistry study, which confirms (after prediction ref 17) an interesting but not revolutionary behavior: the formation of an (O₂)²⁻ dimer in a high pressure ionic compound. This is in line with recent synthesis of metal peroxides such as MgO₂ stable only under pressure. The paper would certainly deserve publication, after further improvement of the form and science, rather in a chemistry or condensed matter journal than in a multidisciplinary journal.

[Authors]: We would like to point out once again that Jang et al. 2017 (ref. 17) "employed DFT + DMFT approaches with treating FeO₂ to **presumably** have O₂²⁻ dimer as a complete analogue of pyrite-structured FeS₂." That is, Jang et al. (2017) did not predict the crystal chemistry of the Py-phase, instead it **assumed** and simply started with the crystal structure of pyrite FeS₂. Our work robustly resolves the debate between previous theoretical studies (e.g. Jang et al. 2017 and Streltsov et al. 2017).

On the other hand, MgO₂ can be stable in the pyrite structure (*Pa3*) at ambient conditions, instead of only at high pressure. It can be commercially purchased from Sigma-Aldrich. In particular, Lobanov et al. (2017, *Stable magnesium peroxide at high pressure, Sci. Rep. 5, 13582*) did not synthesize a metal peroxide MgO₂ in the pyrite structure, but heated up the pyrite-type MgO₂ around 100 GPa and discovered a new high-pressure phase of tetragonal MgO₂.

Data analysis and discussion is still unsatisfactory. Two examples:

(1) L. 137-143, the disappearance of a XES magnetic signal with increasing pressure is described as an indication of a high spin to low spin transition (please correct typo) in FeO₂H_x around 50 GPa. However, the data have been collected on pressure decrease and FeO₂H_x, stable around 100 GPa, decomposes in the same range into magnetic compounds as discussed l. 130-134. No high spin-low spin transition in FeO₂H_x is needed.

[Authors]: The XES measurements have been well-established to study the occurrence of electronic spin transitions in iron at high pressures (Fig. 3). Upon decompression, we experimentally observed that the K_{β}' satellite peak of iron in the Py-phase started to increase around 60-65 GPa where the Py-phase is still metastable and no other phases have appeared based on XRD patterns. Furthermore, the SMS spectra from the sample suggest that there are no magnetic compounds at least at down to 55 GPa (Fig. 2A), where we had already observed an increase in the intensity of the K_{β}' satellite peak (Fig. 3).

Based on the aforementioned observations, we confidently conclude that a low spin to high spin transition of iron was observed with decreasing pressure from 65 to 50 GPa and the metastable Py-phase then decomposed into magnetic compounds below 40-50 GPa.

(2) L. 163-167 a value of O-O bond length of 1.874(1) is reported, based on XRD data analysis. It is presented as an important output of the study. It is IMPOSSIBLE to obtain such accuracy with a Rietvelt refinement of a high pressure XRD powder spectrum, especially for the weakest scatterer of the compound. No Rietvelt fit and goodness parameters are provided. The authors should better discuss the length predicted ab initio, if it is not sensitive to calculations parameters.

[Authors]: The high refinement accuracy for FeO₂ is due to the small number of parameters in the optimization. For Rietveld refinement, Py-FeO₂ has four parameters as variables: the lattice parameter a , atomic coordinate x , and thermal factors for Fe and O. However, as Reviewer #1 pointed out, the weak scattering from O₂ and Fe₂O₃ make Rietveld refinements impossible for all phases. The weak phases were analyzed by *LeBail* fitting. The final refinement factors are $R_I = 0.069$ and $wR_2 = 0.14$. After considering the weak scattering for other phases, we should conservatively claim the O-O distance is approximately 1.87 Å.

Please refer to **Lines 156-162** for changes:

“Due to the importance of the O-O bond length, we collected XRD patterns on a nearly pure FeO₂ sample at 96 GPa, which have been further analyzed by using the Rietveld method for the primary phase of FeO₂ and the LeBail method for the weaker O₂ and Fe₂O₃ phases. The final refinement factors are $R_I = 0.069$ and $wR_2 = 0.14$. The bond length (D_0) for FeO₂ is approximately 1.87(1) Å and the lattice parameter is 4.2855(3) Å with $y = 0.374$ at 96 GPa. Therefore, compared with various peroxides synthesized at ambient pressure³³, the O-O bonding length in the iron superoxide is relatively longer.”

On the other hand, the predicted length *ab initio* is sensitive to calculation parameters. For example, Lu et al. (*Phys. Rev. B* **98**, 054102, 2018) summarized results from various parameters and found either antiferromagnetic or ferromagnetic configurations will yield ~10% longer O-O bonds. Our experiments confirmed that pyrite-FeO₂ and FeO₂Hx are in the non-magnetic state which guided us in then choosing the correct calculation parameters. With employing the experimentally determined magnetic configuration, the O-O bond length calculated from our GGA+ U parameterization is consistent with our XRD experimental results as well as Lu et al.

(2018). What's more, our results from GGA+ U were further verified by HSE calculations that do not depend on the addition of an empirical Hubbard parameter U .

The aforementioned reference and discussion have been added (**Lines 164-181**):

“We further conducted first-principles simulations (see Methods) to construct the charge density landscape at 100 GPa (Fig. 4). Following the Bader charge division scheme, we determined the charge of the Fe atoms are 6.38 e by DFT+ U and 6.32 e by the Heyd-Scuseria-Ernzerhof (HSE) screened hybrid functional (screening variable of 0.15) free of the empirical Hubbard parameter U . Based on our experimental observations that Py-FeO₂H_x is in the non-magnetic state, the O-O bond length values from our GGA+ U calculations are consistent with our XRD experimental results as well as Lu *et al.*³³. We note that the O-O bonding length is sensitive to computational and physical environments and that either antiferromagnetic or ferromagnetic configurations would yield a longer O-O bond by ~10%³³. The valence state of iron is +1.62 (DFT+ U) or +1.68 (HSE) and that of oxygen is -0.81 (DFT+ U) or -0.84 (HSE) for FeO₂ (Table 1). It is known that Bader analysis may underestimate the charge transfer between anions and cations³⁴. Our calculations may not indicate the exact valence values but clearly show charge transfer between anions, which has also been seen in the calculations of archetypal pyrite FeS₂ (ref. 35). In hydrogen-bearing Py-FeO₂H, hydrogen is equally bonded with two nearest oxygen atoms³⁶. As a result, hydrogen is calculated to be +0.64 at 100 GPa. Oxygen atoms adopt electrons from hydrogen and become -1.13 while the valence state of iron remains around +1.63 for the Py-FeO₂H. Therefore, hydrogen only donates electrons to oxygen atoms and has a minimal effect on the valence state of iron.”

My role should not be to correct such elementary mistakes. The paper is not well organized: XRD data are discussed in a paragraph called “Theoretical simulations...”

[Authors]: The discussion of XRD data in this subsection is focused on the O-O bonding length, which is critical for the theoretical simulations. To clarify, the subsection title, “Theoretical simulations on the charge density landscape of FeO₂ and FeO₂H” has been modified to “Oxygen bonding and charge transfer in FeO₂ and FeO₂H.” In addition, the one paragraph has been revised into two paragraphs (**Lines 146-181**).

Reviewer #3 (Remarks to the Author):

In the revised version of the manuscript, the authors addressed thoroughly all the comments and remarks raised during the first critical assessment of the manuscript. The Reviewer is fully satisfied with the current version of the manuscript and recommends its publication in the Nature Communications journal as it is. The work will surely attract the attention of the scientific community in the respective and related fields and will definitely stimulate other studies.

[Authors]: We greatly appreciate Reviewer #3's recommendation for publication in *Nature Communications*.